# Abstract Interpretation of ReLU Neural Networks with Optimizable Polynomial Relaxations

## Abstract

Neural networks have shown to be highly successful in a wide range of applications. However, due to their black box behavior, their applicability can be restricted in safety-critical environments, and additional verification techniques are required. Many state-of-the-art verification approaches use abstract interpretation based on linear overapproximation of the activation functions. Linearly approximating non-linear activation functions clearly incurs loss of precision. One way to overcome this limitation is the utilization of polynomial approximations. A second way shown to improve the obtained bounds is to optimize the slope of the linear relaxations. Combining these insights, we propose a method to enable similar parameter optimization for polynomial relaxations. Given arbitrary polynomials parameterized by their monomial coefficients, we can obtain valid polynomial overapproximations by appropriate upward or downward shifts. Leveraging automatic differentiation, we optimize the choice of the monomial coefficients via gradient-based techniques.

## 1 Introduction

Neural networks (NNs) achieve state of the art performance on many machine learning tasks. However, research by Szegedy et al. (2014) has shown that NNs can exhibit unexpected behavior even in the case of slight input perturbations. Additionally, they comprise a large number of weights and biases that renders them incomprehensible to humans. Despite these problems their impressive performances have led to their use in safety critical areas like airborne collision avoidance (Julian et al., 2016) or autonomous driving (Pan et al., 2020). In order to avoid the severe consequences of failures in these areas of application, formal guarantees for the behavior of NNs are needed. Methods based on MILP (Cheng et al., 2017; Tjeng et al., 2019) or semidefinite programming (Raghunathan et al., 2018; Lan et al., 2022) are capable of providing bounds on the violation of a property via encoding as an optimization problem. Bounds obtained via abstract interpretation-based approaches are the result of propagating sets of inputs through a network. Propagation of symbolic intervals (Wang et al., 2018a;b; Zhang et al., 2018; Singh et al., 2019; Henriksen & Lomuscio, 2020) using convex linear overapproximation of the activation functions has proven to be a promising approach. However, the activation functions are non-linear and the output set of an NN can be non-convex. Consequently, in order to better integrate these methods into the analysis of cyber-physical systems, which often rely on Taylor model arithmetic, researchers have turned to overapproximation techniques involving polynomials (Zhang et al., 2018; Ivanov et al., 2019; 2021; Fatnassi et al., 2022; Huang et al., 2022; Kochdumper et al., 2022). In the context of SIP based on linear overapproximation, multiple valid relaxations of the activation function $\text{ReLU}(x) = \max(0, x)$ have been proposed. Using the insight that the set of valid relaxations can be described by a free (or only interval-constrained) parameter, choosing the value of the parameter via gradient-based optimization of the obtained output bounds led to significantly tighter bounds (Xu et al., 2021). The current state of abstract interpretation based on propagation of polynomials also relies on a small number of proposed ReLU relaxations. In this paper, our main contribution is a parametrization of the set of valid polynomial overapproximations of the ReLU activation function. Similar to the case of propagation of linear symbolic intervals, this parametrization allows for unconstrained optimization to select the parameters of the polynomial ReLU relaxations.

## 2 BACKGROUND

### 2.1 NEURAL NETWORKS

A feed-forward neural network $\mathcal{N}$ comprises an input layer, $L - 1$ hidden layers and an output layer (Goodfellow et al., 2016). The network computes a function $f_{\mathcal{N}} : \mathbb{R}^{d_0} \to \mathbb{R}^{d_L} : \mathbf{x} \mapsto \mathbf{y}$, where the $l$-th layer consists of $d_l$ neurons $\mathbf{n}_l$. Given the weights $W_l \in \mathbb{R}^{d_l \times d_{l-1}}$ and biases $\mathbf{b}_l \in \mathbb{R}^{d_l}$ of layer $l$ and the activation values $\mathbf{n}_{l-1}$ of the previous layer (or the inputs, for $l = 0$, identifying $\mathbf{n}_0$ with the input vector $\mathbf{x}$), the activations of the current layer can be calculated according to

$$\hat{\mathbf{n}}_l = W_l \mathbf{n}_{l-1} + \mathbf{b}_l$$
$$\mathbf{n}_l = \sigma(\hat{\mathbf{n}}_l) \,,$$

where we denote the pre-activation values of layer $l$ by $\hat{\mathbf{n}}_l$ and $\sigma$ is a non-linear activation function. In this paper, we only consider the case of already trained feed-forward NNs with the popular ReLU activation function $\mathrm{ReLU}(x) = \max(0, x)$.

### 2.2 SYMBOLIC INTERVAL PROPAGATION FOR NEURAL NETWORK VERIFICATION

Given a NN $\mathcal{N}$ computing a function $\mathbf{y} = f_{\mathcal{N}}(\mathbf{x})$ on an input set $\mathcal{X}$ and a property $\mathcal{P}(\mathbf{y})$ on the output space $\mathcal{Y} = f_{\mathcal{N}}(\mathcal{X})$ of the NN, the problem of NN verification can be cast as deciding whether the implication

$$\forall \mathbf{x} \in \mathcal{X}, \forall \mathbf{y} \in \mathcal{Y} : \mathbf{y} = f_{\mathcal{N}}(\mathbf{x}) \implies \mathcal{P}(\mathbf{y}) \tag{1}$$

holds or is violated. Although the problem is decidable for feed-forward NNs with ReLU activation functions, it still is NP-complete (Katz et al., 2017).

Therefore, many solvers sacrifice completeness for scalability and use abstract interpretation together with overapproximation of the activation functions to efficiently propagate set representations through the layers of the NN. The property $\mathcal{P}$ then only has to be checked on the set representation obtained after the output layer of the NN.

However, too coarse approximations may not allow proving a desired property. Thus, we can weaken our verification goal and try to obtain an approximation that is as close as possible to the real output region. We therefore search for overapproximations that are as tight as possible.

In its easiest form, hyperrectangles are propagated through the NN via interval arithmetic. As not only the non-linear activation functions, but also the affine layers need to be overapproximated in this setting, large amounts of dependency information are lost, which led to the development of *symbolic interval propagation* (SIP) (Wang et al., 2018b;a). Instead of enclosing the values of neurons $n(\mathbf{x})$, for an input $\mathbf{x}$ of the NN, with concrete bounds $[\underline{n}, \overline{n}] \in \mathbb{R}$, such that $n(\mathbf{x}) \in [\underline{n}, \overline{n}]$, the values of neurons are now enclosed by functions $l(\mathbf{x}) \leq n(\mathbf{x}) \leq u(\mathbf{x})$ (also written as $n(\mathbf{x}) \in [lb(\mathbf{x}), ub(\mathbf{x})]$), where the inequalities hold point-wise for all considered inputs $\mathbf{x} \in \mathcal{X}$ of the NN. Given lower and upper bounding functions $\mathbf{lb}_{l-1}(\mathbf{x})$ and $\mathbf{ub}_{l-1}(\mathbf{x})$ on $\mathbf{n}_{l-1}(\mathbf{x})$, bounding functions for $\hat{\mathbf{n}}_l = W_l \mathbf{n}_{l-1} + \mathbf{b}_l$ can be obtained by calculating

$$\mathbf{lb}_l(\mathbf{x}) = W_l^+ \mathbf{lb}_{l-1}(\mathbf{x}) + W_l^- \mathbf{ub}_{l-1}(\mathbf{x}) + \mathbf{b}_l \tag{2}$$
$$\mathbf{ub}_l(\mathbf{x}) = W_l^+ \mathbf{ub}_{l-1}(\mathbf{x}) + W_l^- \mathbf{lb}_{l-1}(\mathbf{x}) + \mathbf{b}_l \,, \tag{3}$$

where $W_l^+$ and $W_l^-$ denote the positive and negative entries of $W_l$ respectively. After concrete bounds on the input $n \in [\underline{n}, \overline{n}]$ have been obtained from the symbolic bounds on $n$, for example via interval arithmetic (Moore, 1966), they can be used to define lower and upper relaxations $\underline{\mathrm{ReLU}}(n) \leq \mathrm{ReLU}(n) \leq \overline{\mathrm{ReLU}}(n) \, \forall n \in [\underline{n}, \overline{n}]$. This allows for propagation of a symbolic interval $n(\mathbf{x}) \in [lb(\mathbf{x}), ub(\mathbf{x})]$ through the ReLU function according to

$$lb'(\mathbf{x}) = \underline{\mathrm{ReLU}}(lb(\mathbf{x})) \tag{4}$$
$$ub'(\mathbf{x}) = \overline{\mathrm{ReLU}}(ub(\mathbf{x})) \,. \tag{5}$$

Since linear functions can be represented as matrices of their coefficients, which allows for a fast implementation of Equation 2, they are widely used as bounding functions in SIP (Wang et al., 2018a;b; Henriksen & Lomuscio, 2020; Singh et al., 2019; Zhang et al., 2018). As the propagation

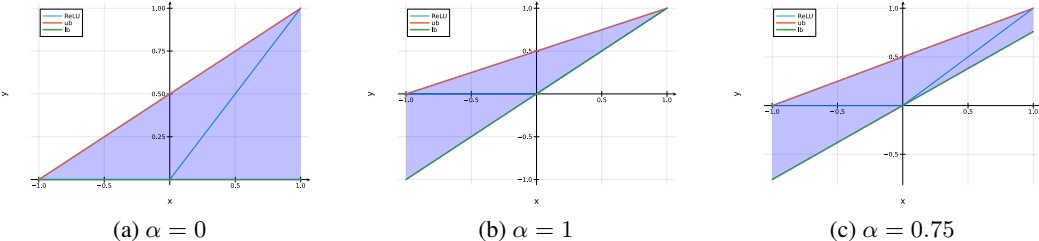

Figure 1: Linear relaxations of the ReLU function for different slopes $\alpha$ of the lower relaxation.

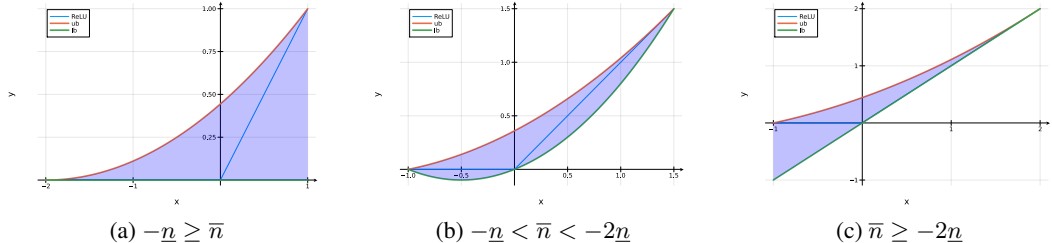

Figure 2: Quadratic relaxations of the ReLU function proposed by Zhang et al. (2018) chosen based on the relationship between the lower and upper concrete bounds $\underline{n} < 0 < \overline{n}$ on the input $n$ of ReLU$(n)$.

of linear symbolic intervals through a ReLU relaxation should again result in a linear symbolic interval, these approaches are restricted to linear ReLU relaxations. If ReLU$(n)$ is *fixed* – i.e. when $\underline{n} \geq 0$ or $\overline{n} \leq 0$ – it already is a linear function over the interval $[\underline{n}, \overline{n}]$ and thus no relaxation is needed. Otherwise we call the neuron *unstable* or *crossing* and we set:

$$\underline{\text{ReLU}}(n) = \alpha n, \ \alpha \in [0,1] \tag{6}$$

$$\overline{\text{ReLU}}(n) = \frac{\overline{n}}{\overline{n} - \underline{n}} n - \frac{\overline{n}\underline{n}}{\overline{n} - \underline{n}} \ . \tag{7}$$

Linear relaxations for different $\alpha$ are shown in Figure 1. Most approaches *statically* choose the value of $\alpha$ based on information locally available at the current neuron. It is set to $\alpha = \frac{\overline{n}}{\overline{n} - \underline{n}}$ to be parallel to the upper relaxation in (Henriksen & Lomuscio, 2020; Singh et al., 2018; Wang et al., 2018b) or as $\alpha = 0$, if $|\underline{n}| \geq \overline{n}$, and $\alpha = 1$, if $|\underline{n}| < \overline{n}$, in (Zhang et al., 2018; Singh et al., 2019) to minimize the volume of the relaxation. However, *any* $\alpha \in [0,1]$ results in a valid lower relaxation. Therefore, given a fixed input set $\mathcal{X}$ and network $\mathcal{N}$, the obtained concrete output bounds can be viewed as a function of the values of $\alpha$ for each individual neuron. This function can then be optimized by varying the parameters $\alpha$ to achieve significantly tighter bounds (Xu et al., 2021).

To overcome the restriction to *linear* ReLU relaxations and increase the potential for tighter relaxations, others have turned to SIP using *polynomials* as bounding functions. The relaxations proposed in the literature again select the coefficients of the relaxation polynomials statically based on the concrete bounds of the input $n \in [\underline{n}, \overline{n}]$ to the activation function. While the methods proposed by Huang et al. (2022) and Kochdumper et al. (2022) are restricted to parallel relaxations, as they build on Taylor models (Makino & Berz, 2003) and polynomial zonotopes (Kochdumper & Althoff, 2021), Zhang et al. (2018) use non-parallel quadratic ReLU relaxations in their tool CROWN

as follows:[1]

$$\underline{\text{ReLU}}(n) = \begin{cases} 0 & , -\underline{n} \geq \overline{n} \\ \frac{\overline{n}}{\overline{n}^2 - \underline{n}\overline{n}} n^2 - \frac{n\overline{n}}{\overline{n}^2 - \underline{n}\overline{n}} & , -\underline{n} < \overline{n} < -2\underline{n} \\ n & , \overline{n} \geq -2\underline{n} \end{cases} \tag{8}$$

$$\overline{\text{ReLU}}(n) = \begin{cases} -\frac{n}{(\underline{n} - \overline{n})^2} n^2 + \frac{n^2 + \overline{n}^2}{(\underline{n} - \overline{n})^2} n - \frac{n\overline{n}^2}{(\underline{n} - \overline{n})^2} & , -\underline{n} \leq \overline{n} \\ \frac{\overline{n}}{(\underline{n} - \overline{n})^2} n^2 - \frac{2n\overline{n}}{(\underline{n} - \overline{n})^2} n + \frac{n^2\overline{n}}{(\underline{n} - \overline{n})^2} & , -\underline{n} > \overline{n} \end{cases} , \tag{9}$$

These quadratic relaxations are visualized in Figure 2. Closed-form parallel quadratic ReLU relaxations are also defined in (Kochdumper et al., 2022). Parallel polynomial relaxations of higher degree can be obtained by regression (Kochdumper et al., 2022) or via approximation by Bernstein polynomials (Huang et al., 2022; Fatnassi et al., 2022). Although parameterizing the ReLU relaxations by $\alpha$ has proven successful in the linear case, finding a similar parameterization for polynomial relaxations has received no attention so far. We will present our solution to this problem in Section 3.

### 2.2.1 BACKWARD SUBSTITUTION

In contrast to this forward propagation of symbolic intervals, several approaches also construct valid linear lower and upper bounding functions by applying substitutions in a backward manner from the output layer to the inputs (Zhang et al., 2018; Singh et al., 2019; Xu et al., 2021). Since computation of a lower bound is analogous, we only review the computation of upper bounds. Given a linear upper bounding function $n_L \leq \mathbf{ub}(\mathbf{n}_l) = \Gamma \mathbf{n}_l + \gamma$ of the output neurons in terms of the neurons in layer $l$ and concrete bounds $\hat{\mathbf{n}}_l \in [\underline{\hat{\mathbf{n}}}_l, \overline{\hat{\mathbf{n}}}_l]$, we can use these bounds to construct a ReLU relaxation and obtain another valid upper bound

$$n_L \leq \Gamma \mathbf{n}_l + \gamma \tag{10}$$

$$= \Gamma \text{ReLU}(\hat{\mathbf{n}}_l) + \gamma \tag{11}$$

$$\leq \Gamma^+ \overline{\text{ReLU}}(\hat{\mathbf{n}}_l) + \Gamma^- \underline{\text{ReLU}}(\hat{\mathbf{n}}_l) + \gamma \tag{12}$$

$$= \Gamma' \hat{\mathbf{n}}_l + \gamma' , \tag{13}$$

this time in terms of the pre-ReLU activation $\mathbf{n}_l$. The substitution in Equation 12 still results in a valid upper bound as we substitute the upper (or lower) ReLU relaxation depending on whether the linear function $\Gamma \mathbf{n}_l + \gamma$ is monotonically increasing (or decreasing) in $n_{l,i}$, which for a linear function only depends on the sign of the respective coefficients in $\Gamma$. Since polynomials need not be monotonic in their input variables, the substitution in Line 12 is in general not possible for polynomial relaxations. Summarizing the terms in Line 12 into the linear function in Line 13 is also only possible for linear ReLU relaxations. In the linear case, using the weights of layer $l$, we can further backsubstitute via

$$n_L \leq \Gamma' \hat{\mathbf{n}}_l + \gamma' = \Gamma' (W_l \mathbf{n}_{l-1} + \mathbf{b}_l) + \gamma' = \Gamma' W_l \mathbf{n}_{l-1} + \Gamma \mathbf{b}_l + \gamma' = \Gamma'' \mathbf{n}_{l-1} + \gamma''$$

to obtain a valid upper bound in terms of the previous layer's outputs $\mathbf{n}_{l-1}$, and repeat this process until the input layer is reached. Concrete bounds $n_L \in [\underline{\mathbf{n}}_L, \overline{\mathbf{n}}_L]$ can then be calculated via interval arithmetic. Bounds calculated via this backsubstitution process are typically significantly tighter than bounds obtained by forward propgation of symbolic intervals, as the dependency problem with respect to intermediate neurons (see (Kern et al., 2022) for a good explanation) is greatly reduced. However, this increased tightness of the bounds is not free: Substitution from layer $l$ backwards requires concrete bounds for all layers $1, \ldots, l - 1$, which – again – have to be calculated using backsubstitution to utilize the improvements in bound tightness. Therefore $\Theta(L^2)$ backsubstitution passes are required to calculate bounds on the output of a $L$-layer NN.

## 3 PARAMETERIZED POLYNOMIAL RELU RELAXATIONS

We use the polynomial representation developed by Kochdumper & Althoff (2021) for sparse polynomial zonotopes. In their approach, important functions for symbolic interval propagation of poly-

---

[1] These relaxations are not described in their publication, but can be found in their tool's implementation at `https://github.com/huanzhang12/CROWN-Robustness-Certification/blob/master/quad_fit.py`

nomials like addition of polynomials, multiplication of polynomials by constant matrices and calculation of concrete lower and upper bounds can be efficiently represented as operations on the matrices of monomial coefficients of the involved polynomials. The same holds true for elementwise application of univariate polynomials. A detailed description is given in Appendix A.1.

Given concrete lower and upper bounds on the input $x \in [\underline{x}, \overline{x}]$ to a ReLU node, and a vector of monomial coefficients $\mathbf{a} = (a_0, a_1, \ldots, a_d)^T$, we can easily construct a valid polynomial upper relaxation $u(x) \geq \text{ReLU}(x) \, \forall x \in [\underline{x}, \overline{x}]$.

For this purpose, we set $p(x) = \sum_{i=0}^{d} a_i x^i$ and shift that polynomial upwards by a sufficiently large $\delta \in \mathbb{R}$ to obtain a valid overapproximation $u(x) = p(x) + \delta$ of the ReLU function. The smallest possible value for $\delta$ can be obtained as the solution to

$$\delta = \max_{x \in [\underline{x}, \overline{x}]} \left( \text{ReLU}(x) - p(x) \right) . \tag{14}$$

Since the ReLU function is piecewise linear, this reduces to finding $\delta_0$ and $\delta_1$ with

$$\delta_0 = \max_{x \in [\underline{x}, 0]} \left( \text{ReLU}(x) - p(x) \right) = \max_{x \in [\underline{x}, 0]} -p(x) \tag{15}$$

$$\delta_1 = \max_{x \in [0, \overline{x}]} \left( \text{ReLU}(x) - p(x) \right) = \max_{x \in [0, \overline{x}]} x - p(x) \tag{16}$$

and setting $\delta = \max(\delta_0, \delta_1)$. Note that solving Equation 16 only requires finding the maxima of *univariate* polynomials.

The above method can be understood as a projection operator taking any vector of monmial coefficients $\mathbf{a}$ and projecting them to the monomial coefficients $\mathbf{a}' = (a'_0, a_1, \ldots, a_d)^T$ of the closest polynomial $u(x) \geq \text{ReLU}(x), \, \forall x \in [\underline{x}, \overline{x}]$ along the dimension of the 0-th coefficient. We therefore obtain valid overapproximations for *any* choice of monomial coefficients $\mathbf{a}$, and thus can use optimization methods to select those coefficients that lead to tight bounds at the output of the NN.

For efficient optimization, we need to be able to calculate the gradients with respect to the given monomial coefficients. Since the maximizers of Equation 16 can be obtained via closed form solutions for polynomials up to degree $d = 4$, it would be possible to differentiate through these computations using automatic differentiation. However, we found that to be inefficient for gradient computation. Instead, we still use an exact approach to find the maximizer $x^*(\mathbf{a})$ of Equation 16 given parameters $\mathbf{a}$. This exact maximizer of $f(x, \mathbf{a}) = \text{ReLU}(x) - p(x \mid \mathbf{a})$ in $[l, u] = [\underline{x}, 0]$ (respectively $[l, u] = [0, \overline{x}]$) is then of course also a local maximizer and thus a fixed point of the projected gradient descent update rule

$$x_{i+1} = \max(l, \min(u, x_i - \eta \nabla_1 f(x_i, \mathbf{a}))) . \tag{17}$$

Using the results of (Blondel et al., 2022), we can then use implicit differentiation as a more efficient way to calculate gradients with respect to $\mathbf{a}$.

Because gradient computation using the view as a fixed point of projected gradient descent is agnostic to the method that calculated the exact maximizer $x^*(\mathbf{a})$, gradients for $\mathbf{a}$ could also be computed by iterative root finding methods (Skowron & Gould, 2012) and therefore enable relaxation by polynomials of degree $d \geq 5$. Our implementation is (currently) restricted to $d = 2$, though.

Construction of valid lower relaxations $l(x) \leq \text{ReLU}(x) \, \forall x \in [\underline{x}, \overline{x}]$ is straightforward. We simply set $l(x) = p(x) + \delta$, but instead of a maximization, we solve the minimization problem

$$\delta = \min_{x \in [\underline{x}, \overline{x}]} \text{ReLU}(x) - p(x) . \tag{18}$$

If we use different monomial coefficients $\mathbf{a}_l$ and $\mathbf{a}_u$ as a basis for our construction, our method is able to produce non-parallel lower and upper relaxations. However, it can also be incorporated into propagation techniques that require parallel relaxations, as those presented by Kochdumper & Althoff (2021) and Kochdumper et al. (2022). In this case, we have to use the same monomial coefficients for the construction of the lower and the upper relaxation.

## 3.1 OPTIMIZATION OF OUTPUT BOUNDS

Given a concrete vector $\theta = \{(\mathbf{a}_l^k, \mathbf{a}_u^k) \mid k = 1, ..., L\}$ containing all required monomial coefficients for the unstable neurons in a network, concrete lower and upper bounds on the network's outputs

$\underline{\mathbf{lb}}, \overline{\mathbf{ub}}$ can be computed via

$$\mathbf{lb}(\mathbf{x}), \mathbf{ub}(\mathbf{x}) = \textsc{Propagate}(\mathcal{X} \mid \theta) \tag{19}$$

$$\underline{\mathbf{lb}}, \overline{\mathbf{lb}} = \textsc{Concretize}(\mathbf{lb}(\mathbf{x})) \tag{20}$$

$$\underline{\mathbf{ub}}, \overline{\mathbf{ub}} = \textsc{Concretize}(\mathbf{ub}(\mathbf{x})) \,. \tag{21}$$

As PROPAGATE depends on the parameters $\theta$ and all functions used to obtain $\underline{\mathbf{lb}}$ and $\overline{\mathbf{ub}}$ are differentiable, we can use gradient-based optimization techniques to optimize the tightness of the obtained concrete lower and upper bounds. Since the number of unstable neurons in a network can be large and each unstable neuron requires $2d + 2$ monomial coefficients for forward overapproximation by non-parallel polynomial lower and upper relaxations of degree $d$, we need to perform optimization over a high dimensional parameter space. In this setting, it is crucial to utilize that due to our parametrization of the polynomial ReLU relaxations, any value for $\theta$ leads to valid lower and upper bounds. Therefore, we can treat the problem as an unconstrained optimization problem, which enables us to profit from the efficiency of unconstrained gradient-based optimisation.

The loss function $\mathcal{L}$ can be any function of the concrete (or even the symbolic) output bounds. We chose to minimize the sum of the widths of the concrete output bounds by setting $\mathcal{L} = \sum_{i=1}^{n_L} \overline{ub}_i - \underline{lb}_i$ to obtain tight concrete bounds.

Depending on the number of unstable neurons overapproximated by polynomial relaxations, we may need to use norm-clipping of the gradients to avoid numerical difficulties caused by large gradients.

We present two instantiations of the function PROPAGATE($\mathcal{X} \mid \theta$): Forward propagation of symbolic intervals as introduced by Wang et al. (2018b), but with polynomial ReLU relaxations, as well as integrating polynomial ReLU relaxations into the backsubstitution approach presented by Xu et al. (2021) and reviewed in Section 2.2.1. Since polynomials are not necessarily monotonic in their input variables, the substitution procedure described in equations 10-13 is in general not possible, if we replace the linear upper bounding function $\mathbf{n}_L \leq \Gamma \mathbf{n}_l + \gamma$ by a polynomial bounding function $\mathbf{n}_L \leq ub(\mathbf{n}_l)$. However, we can calculate a polynomial symbolic interval $\mathbf{n}_1 \in [\mathbf{lb}(\mathbf{x}), \mathbf{ub}(\mathbf{x})]$ via forward propagation and use linear backsubstitution from layer $l \geq 2$ only down to the first layer to obtain a linear upper bounding function in terms of $\mathbf{n}_1$ and then substitute

$$\mathbf{n}_l \leq \Gamma \mathbf{n}_1 + \gamma \tag{22}$$

$$\leq \Gamma^+ \mathbf{ub}(\mathbf{x}) + \Gamma^- \mathbf{lb}(\mathbf{x}) + \gamma \tag{23}$$

as the linear function is of course monotonic. The resulting upper bounding function is a polynomial and as we have to perform one backsubstitution pass from layer $l$ down to layer 1 for each layer $2 \leq l \leq L$ of the NN, each of the resulting symbolic bounding functions is a polynomial and the bounds for layer $l$ benefit from the tighter polynomial relaxation of the first hidden layer.

An example for forward propagation of optimizable polynomial symbolic intervals can be found in Appendix A.2.

## 4 EVALUATION

**Benchmarks.** Since the number of monomial terms of a polynomial grows rapidly with the number of unfixed input dimensions, we need to restrict our evaluation to verification problems with a small or intermediate number of unfixed inputs. We therefore evaluated our approach on the two fully connected networks with ReLU activation functions, ACAX Xu and MNIST, from the annual NN competitions VNN-COMP[2] (Brix et al., 2023). The ACAS Xu benchmark set was published by Katz et al. (2017) and consists of NNs that were trained to provide navigation advisories on board of aeroplanes as part of the Airborne Collision Avoidance System for unmanned aircraft. Each of the 45 NNs has a small input dimension of only 5 input neurons, 6 fully connected layers of 50 neurons each, and 5 output neurons representing scores for the different navigation advisories. All of the properties define a hyperrectangle of valid values for the input space and an appropriate property on the output space of the NN. The MNIST benchmark set deals with classification of handwritten digits. It consists of 3 fully-connected networks with 2, 4 or 6 layers of 256 neurons each. The input

---

[2]https://github.com/stanleybak/vnncomp2021

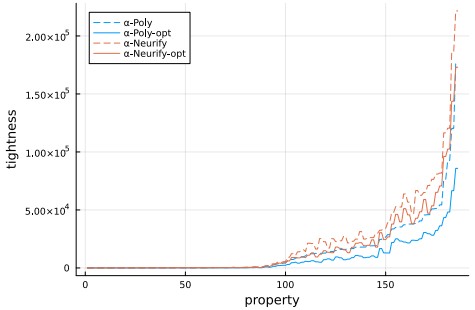
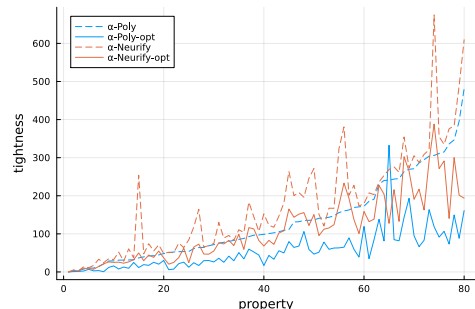

(a) Tightness of output bounds for all ACAS-Xu properties

(b) Tightness of output bounds for the first 80 properties

Figure 3: Comparison of the tightness (sum of the widths) of the output bounds for each of the ACAS-Xu properties sorted by the tightness after initialization of $\alpha$-POLY to $\alpha$-NEURIFY.

dimension is 784 for the greyscale level of 28x28 pixels. We use the original three fully connected NNs, but instead of considering small $L_\infty$-perturbations to all 784 pixels of the 15 input images, we consider adversarial patches of varying sizes up to 50 pixels, where each unfixed pixel can admit the whole range of grayscale values.

**Tools.** We evaluate the performance of our optimizable polynomial relaxations instantiated with forward propagation of polynomials – which we call $\alpha$-POLY – and our second instantiation – which we call POLYCROWN – and which is based on inclusion of polynomial relaxations in the first layer of the backsubstitution procedure $\alpha$-CROWN (Xu et al., 2021) with optimizable linear relaxations. Our Julia implementation can be found at [3].

We compare our approach to $\alpha$-NEURIFY – our implementation of NEURIFY (Wang et al., 2018b), where we included optimization of the linear relaxations – and our implementation of $\alpha$-CROWN (Xu et al., 2021) [4].

We further compare to forward propagation of zonotopes and polynomial zonotopes (Kochdumper et al., 2022). As suggested by Kochdumper et al. (2022), we use polynomial relaxations for the first two layers of an NN and then switch to linear relaxations.

**ACAS Xu.** For each of the 186 verification instances, we propagate the associated input set through the NN and record the sum of the widths of the output bounds obtained by using static ReLU relaxations (which are also used to initialize the optimizable relaxations) and the tightness of the output bounds after optimization. Optimization for an instance is stopped as soon as a timeout of 5 minutes, a predefined number of steps is reached or the objective did not improve for 50 steps. We abort optimization for $\alpha$-CROWN and POLYCROWN after at most 5000 steps and for $\alpha$-NEURIFY after at most 40000 steps.

Our results on forward propagation of symbolic intervals shows that forward propagation of polynomials is – at least when output bounds are large – already tighter in most cases than the bounds obtained via optimized linear relaxations as can be seen on the right of Figure 3a. Even when the output bounds are tighter, we observe that the initial values using polynomial relaxations as well as the values after optimization are better than the initial and optimized values obtained via linear relaxations. as is evident from Figure 3b.

Although not as pronounced as in the case of forward propagation, we see from Figure 4 that the initial output bounds obtained via POLYCROWN are mostly tighter than the initial bounds

---

[3]Anonymous OSF-link for double blind review: `https://osf.io/p23ga/?view_only=85448974a7e34dd3bedb4c3c8d86987d`

[4]We chose to reimplement $\alpha$-CROWN due to difficulties with the execution of single benchmark instances and for better accessiblity to data during the optimization process. We confirmred via testing that our implementation produces the same bounds as the original implementation of $\alpha$-CROWN, when the parameter `shared_alpha=True` is set.

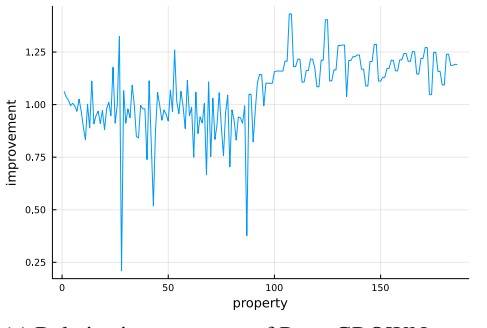

(a) Relative improvement of POLYCROWN vs $\alpha$-CROWN

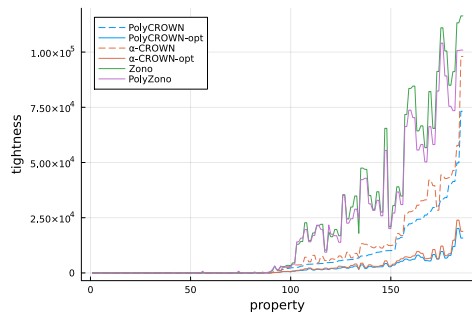

(b) Tightness of output bounds

Figure 4: Comparison of the tightness (sum of the widths) of the output bounds for each of the ACAS-Xu properties sorted by the tightness after initialization of POLYCROWN to $\alpha$-CROWN as well as propagation of zonotopes and polynomial zonotopes.

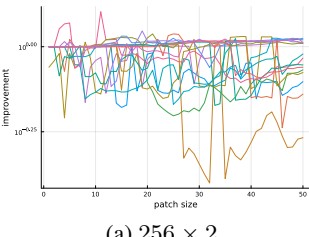

(a) $256 \times 2$

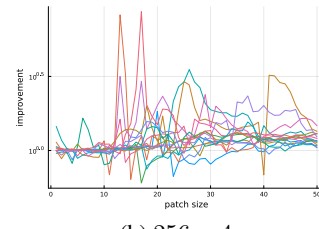

(b) $256 \times 4$

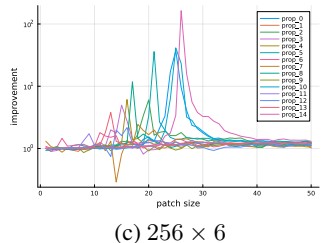

(c) $256 \times 6$

Figure 5: Relative improvement of POLYCROWN vs $\alpha$-CROWN after initialization for different MNIST networks. Note the logarithmic scale! For property 14 the initial POLYCROWN bounds are 163 times tighter than the initial bounds obtained using $\alpha$-CROWN.

of $\alpha$-CROWN for large output bounds and significantly outperform the bounds obtained by propagation of zonotopes or polynomial zonotopes. A closer look at the relative improvement $\mathcal{L}_{\alpha-\text{CROWN}}/\mathcal{L}_{\text{POLYCROWN}}$ as shown in Figure 4a reveals that the bounds obtained by POLYCROWN are on average 18% tighter, when $\mathcal{L}_{\alpha-\text{CROWN}\geq 100}$, while being on average 6% worse on the remaining instances.

**MNIST.** We compare the performance of POLYCROWN to $\alpha$-CROWN for all three networks and all 15 images in the benchmark. However, as a single propagation of an input set through a network is very efficient compared to a whole optimization run, we evaluate bounds tightness after initialization for all patch sizes between 1 and 50 pixels, while only considering patch sizes in $\{15, 20, 25\}$ for the evaluation of optimized bounds.

The results after initialization again show that the initial bounds obtained by POLYCROWN can significantly outperform the bounds obtained by linear relaxations in many cases. While there is almost no improvement for the smaller $256 \times 2$ network, bound tightness for the $256 \times 4$ network mostly improves whereas a relative improvement $\mathcal{L}_{\alpha-\text{CROWN}}/\mathcal{L}_{\text{POLYCROWN}}$ of up to $163\times$ can be observed for the output bounds on the largest $256 \times 6$ network. We believe that the performance of linear and polynomial relaxations is roughly on a similar scale, when most of a network's neurons are fixed – when the adversarial patches only have a small number of unfixed pixels in the left Figure 5c – and similarly, when the majority of the neurons are crossing – when the patches consist of a very large number of unfixed pixels in the right of Figure 5c. Large improvements seem to happen between these two extremes, when the polynomial relaxations are able to compute bounds in the intermediate layers that are tight enough for some neurons to still be fixed, while the lack of tighness of the linear relaxations forces $\alpha$-CROWN to start overapproximating these neurons, leading to coarse output bounds.

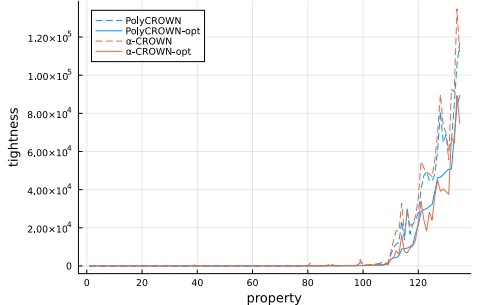

(a) Tightness of output bounds for all MNIST properties

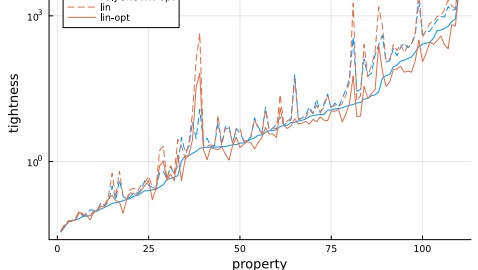

(b) Tightness of output bounds for the first 110 properties. Note the logarithmic scale!

Figure 6: Comparison of the tightness (sum of the widths) of the output bounds for each of the MNIST properties sorted by the tightness after initialization of POLYCROWN to $\alpha$-CROWN.

Table 1: Comparison of the tightness (sum of the widths, $\ell_1$ norm) of the output bounds after a timeout of one hour for the 5 MNIST properties where POLYCROWN performed worst compared to $\alpha$-CROWN after a 5-minute timeout.

| POLYCROWN | POLYCROWN-opt | $\alpha$-CROWN | $\alpha$-CROWN-opt |
|---|---|---|---|
| 1545.61 | 41.15 | 2106.62 | **23.21** |
| 3.13 | 0.4032 | 2.08 | **0.4014** |
| 408.09 | **51.39** | 355.91 | 59.00 |
| 942.27 | **46.09** | 1343.97 | 71.86 |
| 260.83 | 16.88 | 264.85 | **15.84** |

Counterintuitively, optimization of the output bounds using a timeout of 5 minutes shows that the tightness of the optimized linear bounds is oftentimes better than the results for optimized polynomial bounds. This can be explained as the cost per iteration scales quadratically in the number of unfixed input dimensions for POLYCROWN with quadratic ReLU relaxations in contrast to the linear relaxations of $\alpha$-CROWN. While this difference in runtime was not significant in the ACAS-Xu benchmark, in the MNIST benchmark, a single propagation of POLYCROWN was on average $5.2\times$ slower than $\alpha$-CROWN for patches of size 15 and on average $7.6\times$ slower for patches of size 25. As expected, the iteration speed of $\alpha$-CROWN did not change significantly with increasing patch size.

To further investigate the potential of optimizable polynomial relaxations, we select the 5 properties where our approach performed worst compared to $\alpha$-CROWN and repeat the experiments with a longer timeout of one hour. The results summarized in Table 1 show that POLYCROWN was able to come significantly closer to $\alpha$-CROWN on three of five instances and even significantly surpass $\alpha$-CROWN on the remaining two. Progress of the optimization runs is shown in Figure 9 in Appendix A.3. Note that $\alpha$-CROWN achieves a significantly higher iteration count than POLYCROWN.

## 5 CONCLUSION

Parameterized ReLU relaxations have been proposed for abstract interpretation of NNs with linear symbolic intervals. These relaxations have allowed for the calculation of tighter output bounds by optimization of their parameters. In this paper, we proposed the first parameterized ReLU relaxation for propagation of polynomial symbolic intervals by taking arbitrary polynomials and shifting them up or down by a sufficient amout to obtain valid ReLU overapproximations. We demonstrate in our evaluation that our optimized polynomial relaxations can achieve significantly tighter output bounds for many verification problems.

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

## A    APPENDIX

### A.1    REPRESENTATION OF POLYNOMIALS

The efficiency of SIP with polynomial bounds is highly dependent on an efficient implementation of addition of polynomials, multiplication of a polynomial by constant matrices, as well as elementwise application of univariate polynomials and fast, but precise calculation of concrete lower and upper bounds. Therefore, we utilize the polynomial representation developed for sparse polynomial zonotopes by Kochdumper & Althoff (2021). In this setting, the tuple

$$p(\mathbf{x}) = \langle G, E, \mathbf{id} \rangle , \tag{24}$$

where $G \in \mathbb{R}^{n \times m}, E \in \mathbb{N}^{d \times m}$ and $\mathbf{id} \in \mathbb{N}^d$, represents a multidimensional polynomial $p : \mathbb{R}^d \to \mathbb{R}^n$

$$p(\mathbf{x}) = \sum_{i=1}^m \left( \prod_{k=1}^d x_k^{E_{ki}} \right) G_i \tag{25}$$

with $m$ monomial terms. In the remainder of this paper, we sometimes just write *polynomial* instead of *multidimensional polynomial* where it is clear from the context. The monomial coefficients for the $i$-th monomial are stored in the $i$-th column $G_i$ of the generator matrix and the corresponding powers of the $d$ variables are stored in the $i$-th column $E_i$ of the exponent matrix. An identifier for the $k$-th variable is stored in $\mathbf{id}_k$.

For example, the tuple

$$\left\langle \begin{pmatrix} 5 & 4 & 3 & 0 \\ 2 & 3 & 0 & -7 \end{pmatrix}, \begin{pmatrix} 0 & 1 & 2 & 0 \\ 0 & 1 & 0 & 2 \end{pmatrix}, \begin{pmatrix} 1 \\ 2 \end{pmatrix} \right\rangle \tag{26}$$

is a representation of the polynomial

$$p(x_1, x_2) = \begin{pmatrix} 5 \\ 2 \end{pmatrix} + \begin{pmatrix} 4 \\ 3 \end{pmatrix} x_1 x_2 + \begin{pmatrix} 3 \\ 0 \end{pmatrix} x_1^2 + \begin{pmatrix} 0 \\ -7 \end{pmatrix} x_2^2 . \tag{27}$$

Given polynomials $p : \mathbb{R}^d \to \mathbb{R}^n, p(\mathbf{x}) = \langle G_p, E_p, \mathbf{id} \rangle$ and $q : \mathbb{R}^d \to \mathbb{R}^n, q(\mathbf{x}) = \langle G_q, E_q, \mathbf{id} \rangle$ depending on the same input variables $\mathbf{x}$ and a matrix $A \in \mathbb{R}^{t \times n}$, a representation of the addition of two polynomials is given as

$$p(\mathbf{x}) + q(\mathbf{x}) = \langle [G_p \quad G_q], [E_p \quad E_q], \mathbf{id} \rangle . \tag{28}$$

Duplicate monomials – i.e. duplicate columns in $[E_p\ E_q]$ – can be summarized by only keeping one of these columns along with the sum of the corresponding columns in the new generator matrix $[G_p\ G_q]$. If both polynomials share the same exponent matrix (i.e. $E_p = E_q$), there are no redundant monomials and the expression reduces to

$$p(\mathbf{x}) + q(\mathbf{x}) = \langle G_p + G_q, E, \mathbf{id} \rangle \,, \tag{29}$$

thus saving the cost of summarizing duplicate monomials.

The linear map also admits an efficient representation by multiplication with the generator matrix $G_p$:

$$A\,p(\mathbf{x}) = \langle A\,G_p, E_p, \mathbf{id} \rangle \tag{30}$$

For ease of notation, we now let $p(\mathbf{x}) = \langle G, E, \mathbf{id} \rangle$ and additionally consider a polynomial $f(\mathbf{y}):\mathbb{R}^n \to \mathbb{R}^n$. The component-wise polynomial map is defined as the polynomial $r(\mathbf{x}):\mathbb{R}^d \to \mathbb{R}^n$ where $r_i(\mathbf{x}) = f_i(p_i(\mathbf{x}))$, $\forall i \in [n]$. In that case, all $f_i(x) = \sum_{j=0}^{k} a_{ij} x^j$ are *univariate* polynomials, each of which can be described by a vector $(a_{i0}, a_{i1}, \dots, a_{ik})$ of monomial coefficients. We can then write $f_i(p_i(\mathbf{x})) = \sum_{j=0}^{k} a_{ij} p_i(\mathbf{x})^j$. Since addition of polynomials was already introduced above, we only need to consider the operation of raising a polynomial to a given power $j$, which is given by

$$p(\mathbf{x})^j = \left\langle \hat{G}, \hat{E}, \mathbf{id} \right\rangle \,, \tag{31}$$

with

$$\hat{G} = \left[ \hat{G}_1\ \hat{G}_2 \dots \hat{G}_{\binom{j+m-1}{m-1}} \right] \quad \hat{E} = \left[ \hat{E}_1\ \hat{E}_2 \dots \hat{E}_{\binom{j+m-1}{m-1}} \right]$$
$$\hat{G}_i = \binom{j}{\alpha_1, \alpha_2, \dots, \alpha_m} \prod_{l=1}^{m} G_{il}^{\alpha_l} \quad \hat{E}_i = \sum_{l=1}^{m} \alpha_l E_l \,,$$

where $\binom{j}{\alpha_1, \alpha_2, \dots, \alpha_m}$ is the multinomial coefficient and the index $1 \le i \le \binom{j+m-1}{m-1}$ enumerates the possible combinations of $\alpha_1 + \alpha_2 + \dots + \alpha_m = j$ such that $\alpha \ge 0$. Note that this number is in $\Theta(m^j)$ for fixed $j \in \mathbb{N}$.

The validity of this result can be seen from expansion of $p(\mathbf{x})^j$ using the multinomial theorem:

$$p_i(\mathbf{x})^j = \left( \sum_{l=1}^{m} \left( \prod_{k=1}^{d} x_k^{E_{kl}} \right) G_{il} \right)^j \tag{32}$$

$$= \sum_{\substack{\alpha_1+\alpha_2+\dots+\alpha_m=j \\ \alpha \ge 0}} \binom{j}{\alpha_1, \alpha_2, \dots, \alpha_m} \prod_{l=1}^{m} \left( \left( \prod_{k=1}^{d} x_k^{E_{kl}} \right)^{\alpha_l} G_{il}^{\alpha_l} \right) \tag{33}$$

$$= \sum_{\substack{\alpha_1+\alpha_2+\dots+\alpha_m=j \\ \alpha \ge 0}} \binom{j}{\alpha_1, \alpha_2, \dots, \alpha_m} \prod_{l=1}^{m} \left( x_k^{\left( \sum_{l=1}^{m} \alpha_l E_{kl} \right)} \right) \prod_{l=1}^{m} G_{il}^{\alpha_l} \,. \tag{34}$$

Concrete lower and upper bounds $\underline{p}$ and $\overline{p}$ of a polynomial $p(\mathbf{x}) = \langle G, E, \mathbf{id} \rangle$ can be efficiently computed via interval arithmetic. To further increase the efficiency and also the tightness of the bounds calculation, we first normalize the input variables to $\mathbf{x} \in [-1, 1]^d$ (Althoff et al., 2018). and then obtain:

$$\underline{p} = G^+ \underline{E} + G^- \overline{E} \qquad \overline{p} = G^+ \overline{E} + G^- \underline{E} \tag{35}$$

with

$$\underline{E} = \begin{pmatrix} \underline{e_1} \\ \vdots \\ \underline{e_m} \end{pmatrix} \quad \underline{e_i} = \begin{cases} 1, & E_{ki} = 0,\ \forall k \text{ (constant monomial)} \\ 0, & E_{ki} \mod 2 = 0,\ \forall k \\ -1, & \text{otherwise} \end{cases} \quad \overline{E} = \begin{pmatrix} 1 \\ \vdots \\ 1 \end{pmatrix} \,.$$

Note that normalization only has to be done once before propagating the input set through the network. When the number of monomial terms $m$ of a polynomial $p(\mathbf{x})$ exceeds a predefined threshold $n_{terms}$, we can compute concrete bounds $\underline{\mathbf{m}}, \overline{\mathbf{m}}$ for the $m - n_{terms}$ monomials whose columns $G_i$ in the generator matrix have the smallest $L_2$-norm, remove these monomials and add $\underline{\mathbf{m}}\ (\overline{\mathbf{m}})$ to the constant monomial to get a valid lower (upper) bounding polyomial $p'(\mathbf{x})$ with less monomial terms.

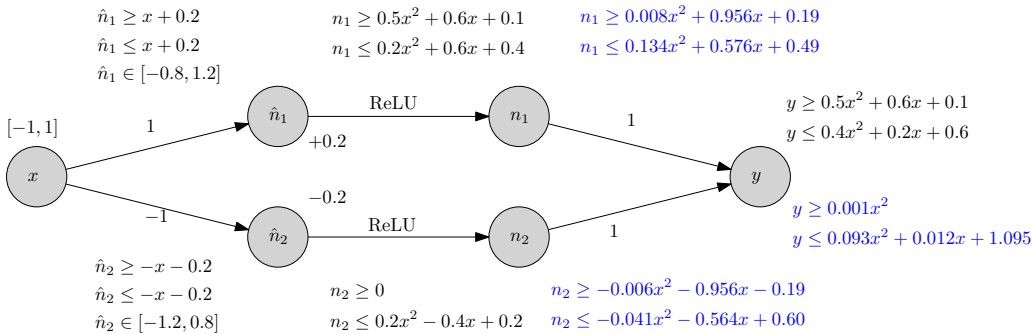

Figure 7: Propagation of polynomials through an example network. Results of the static quadratic relaxations used in CROWN (Zhang et al., 2018) are shown in black. Differences when propagation is optimized to minimize the interval enclosure of the output are shown in blue. All coefficients are rounded to three significant digits.

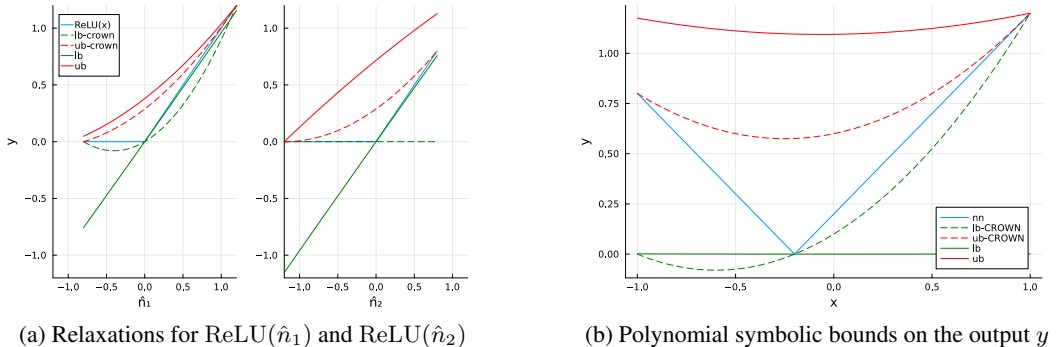

(a) Relaxations for $\mathrm{ReLU}(\hat{n}_1)$ and $\mathrm{ReLU}(\hat{n}_2)$    (b) Polynomial symbolic bounds on the output $y$

Figure 8: Polynomial ReLU relaxations and symbolic output bounds for our example network. The result of the static relaxations are shown as dashed lines. The results after optimizing the relaxations giving tighter output bounds are shown as solid lines.

## A.2 Example

To illustrate the positive effect of the proposed optimizable ReLU relaxations, we show how they can be used to obtain better bounds on the output neuron of the example network shown in Figure 7 when compared to forward propagation of symbolic intervals using the static relaxations used in CROWN (Zhang et al., 2018). The symbolic intervals for $\hat{n}_1$ and $\hat{n}_2$ are calculated according to Equation 2 as

$$\begin{pmatrix} \hat{n}_1 \\ \hat{n}_2 \end{pmatrix} \geq W^+ x + W^- x + \mathbf{b} = \begin{pmatrix} 1 \\ 0 \end{pmatrix} x + \begin{pmatrix} 0 \\ -1 \end{pmatrix} x + \begin{pmatrix} 0.2 \\ -0.2 \end{pmatrix}$$

where the lower and upper bounding functions of $x$ are just $x$, since no overapproximation was necessary yet. The upper bound is calculated similarly.

We then use interval arithmetic to obtain concrete bounds $\hat{n}_1 \in [-0.8, 1.2]$ and $\hat{n}_2 \in [-1.2, 0.8]$ on the pre-ReLU activations.

Using these concrete bounds, we can apply the static ReLU-relaxations given in Equations 8 and 9 proposed by Zhang et al. (2018) to obtain

$$\begin{aligned} \underline{\mathrm{ReLU}(\hat{n}_1)} &= 0.5\hat{n}_1^2 + 0.4\hat{n}_1 & \overline{\mathrm{ReLU}(\hat{n}_1)} &= 0.2\hat{n}_1^2 + 0.52\hat{n}_1 + 0.2 \\ \underline{\mathrm{ReLU}(\hat{n}_2)} &= 0 & \overline{\mathrm{ReLU}}(\hat{n}_2) &= 0.2\hat{n}_2^2 + 0.48\hat{n}_2 + 0.2 \,, \end{aligned}$$

which are represented by the dashed lines in Figure 8a.

Symbolic bounds for $\hat{n}_1, \hat{n}_2$ can then be calculated according to Equation 4. We only show the process for $\hat{n}_1$:

$$n_1 \geq \underline{\text{ReLU}}(x + 0.2) = 0.5(x + 0.2)^2 + 0.4(x + 0.2) = 0.5x^2 + 0.6x + 0.1$$
$$n_1 \leq \overline{\text{ReLU}}(x + 0.2) = 0.2(x + 0.2)^2 + 0.52(x + 0.2) + 0.2 = 0.2x^2 + 0.6x + 0.4 \ .$$

Finally, symbolic bounds for the output neuron $y$ are then again calculated via the formula in Equation 2 as

$$y \geq (1\ 1) \begin{pmatrix} 0.5x^2 + 0.6x + 0.1 \\ 0 \end{pmatrix} + (0\ 0) \begin{pmatrix} 0.2x^2 + 0.6x + 0.4 \\ 0.2x^2 - 0.4x + 0.2 \end{pmatrix} = 0.5x^2 + 0.6x + 0.1 \tag{36}$$

$$y \leq (1\ 1) \begin{pmatrix} 0.2x^2 + 0.6x + 0.4 \\ 0.2x^2 - 0.4x + 0.2 \end{pmatrix} + (0\ 0) \begin{pmatrix} 0.5x^2 + 0.6x + 0.1 \\ 0 \end{pmatrix} = 0.4x^2 + 0.2x + 0.6 \ , \tag{37}$$

which are shown as dashed lines in Figure 8b. Using interval arithmetic, we can obtain concrete bounds of $y \in [-0.5, 1.2]$. Exact optimization yields a tighter interval enclosure of $y \in [-0.08, 1.2]$.

When our parameterized ReLU relaxations are used to optimize the monomial coefficients of the ReLU relaxations in order to obtain tighter concrete bounds on $y$, we can indeed improve the obtained interval enclosure to $y \in [0, 1.2]$. The resulting symbolic bounds are shown as solid lines in Figure 8b. Due to our flexible parameterization, the optimization process was able to find the ReLU relaxations (shown in Figure 8a as solid lines) that allow the symbolic lower bounds of $n_1$ and $n_2$ to almost completely cancel out, when calculating the symbolic lower bound of $y$.

### A.3 FURTHER EVALUATION RESULTS

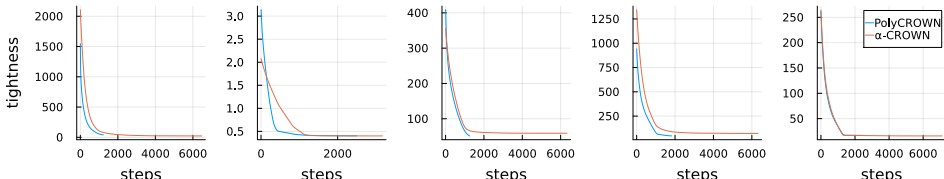

Figure 9: Comparison of the tightness (sum of the widths) of the output bounds after a timeout of one hour for the 5 MNIST properties where POLYCROWN performed worst compared to $\alpha$-CROWN.

