# OpenReview forum: "Abstract Interpretation of ReLU Neural Networks with Optimizable Polynomial Relaxations"
_ICLR.cc/2024/Conference — Submitted to ICLR 2024_

### Official Review · Reviewer_yS99 · 2023-10-26

**Soundness:** 3 good
**Presentation:** 2 fair
**Contribution:** 1 poor
**Rating:** 3
**Confidence:** 4

**Summary:**

The paper proposes to parameterize the univariate polynomial (order 2) ReLU transformers for multidimensional polynomial Zonotope abstractions employed in Neural Network Verification using freely optimizable coefficients. To still obtain a sound abstraction, they shift the resulting polynomials up or down by the maximum soundness violation. They introduce two NN verification methods, $\alpha$-Poly and PolyCROWN, based on full or partial (1 layer) forward propagation of the polynomial Zonotopes, respectively, combined with linear backsubstitution in the latter case. They empirically demonstrate for very low dimensional (d=5) inputs, that optimizing transformer parameters for forward propagation of polynomial Zonotopes ($\alpha$-Poly) yields better tightness than for linear Zonotopes (without additional error terms). For the backsubstitution-based approaches (PolyCROWN and $\alpha$-CROWN, which seem to obtain better overall tightness), this improvement is significantly smaller. In a slightly higher dimensional setting ($d \in [15,25]$), a purely linear approach seems to perform better for moderate timeouts (5 min) and comparable for large timeouts (1h).

**Strengths:**

* The tackled issue of (certified) adversarial robustness is of high importance.
* Their approach of ensuring abstraction soundness by translation of the polynomial is simple and elegant, permitting unconstrained optimization of coefficients.
* The paper demonstrates that optimizing transformer parameterization can improve resulting bound tightness in multiple settings.

**Weaknesses:**

* Key related work on NN verification is missing. Both multi-neuron constraints [1,2,3,4] and a Branch-and-Bound based approaches [5,6,7,4] have dominated the international verification of neural networks competitions (VNN-COM) [8,9] and should be the benchmark for any NN verification method. Specifically, their baseline $\alpha$-CROWN has been superseeded twice by $\beta$-CROWN [6] and then GCP [4].
* The scalability of the proposed method is extremely limited both in terms of model depth and input dimensionality, with the novel optimization being only applied to at most 25 free input dimensions (in contrast, most popular benchmarks have between 784 (MNIST) and 12k (TinyImageNet) free input dimensions [8,9]). Even there, $\alpha$-CROWN consistently performs better for a 5-minute timeout per property, with PolyCROWN only demonstrating an advantage at a 1-hour timeout, where linear bound optimization has become inefficient (see Figure 9). A comparison to BaB-based methods would be essential to establish the promise of the proposed method.
* The fairness of the empirical comparison is unclear as the authors re-implement baseline methods and only verify their correctness but not performance. Additionally, an optimization iteration count based termination condition is used for the only setting (ACAS Xu) where the proposed method shows performance improvement (using different thresholds per method and none at all for their $\alpha$-Poly). However, the proposed method incurs much higher per iteration costs (exponential in network depth for  $\alpha$-Poly) than the baseline (linear in network depth for $\alpha$-Neurify).
* The only setting where the proposed method seems to notably improve bound tightness (forward propagation ($\alpha$-Poly) for ACAS Xu), seems to generally yield much looser bounds than the (partial) backsubsitution based method (Poly/$\alpha$-CROWN), casting doubt on the relevance of these improvements (compare Figure 3a and 4b).
* The presentation could be greatly improved (see comments below).

**References**
1) Singh et al. "Beyond the single neuron convex barrier for neural network certification."  NeurIPS 2019
2) Müller et al. "PRIMA: general and precise neural network certification via scalable convex hull approximations." POPL 2022
3) Tjandraatmadja et al. "The convex relaxation barrier, revisited: Tightened single-neuron relaxations for neural network verification." NeurIPS 2020
4) Zhang et al. "General cutting planes for bound-propagation-based neural network verification." NeurIPS 2022
5) Bunel et al. "Branch and bound for piecewise linear neural network verification." JMLR 2020
6) Wang et al. "Beta-crown: Efficient bound propagation with per-neuron split constraints for neural network robustness verification." NeurIPS 2021
7) Ferrari et al. "Complete verification via multi-neuron relaxation guided branch-and-bound." ICLR 2022
8) Brix et al. "First three years of the international verification of neural networks competition (VNN-COMP)." ISTTT 2023
9) Müller et al. "The third international verification of neural networks competition (VNN-COMP 2022): summary and results." arXiv 2022

**Questions:**

### Questions
1) How frequently is the number of optimization steps used to terminate optimization for AcasXu? Given the much larger computational cost of a single PolyCROWN/$\alpha$-Poly compared to $\alpha$-CROWN/$\alpha$-Neurify step, this termination condition might lead to a biased comparison.
2) Can you compare to the original implementation of $\alpha$-CROWN and more importantly a BaB based method such as $\beta$-CROWN or MN-BaB, which should be much more effective given the long timeouts used here?
3) Can you report results on MNIST per number of free parameters? Further, can you extend this comparison to larger patches (up to all 784 pixels, with a reduced perturbation magnitude), given that polynomial abstractions are only propagated a single layer in this setting? How does $\alpha$-Poly scale in the MNIST setting (for small patch sizes)?
4) Can you report per iteration times for $\alpha$-Poly and $\alpha$-Neurify in the ACAS Xu (and ideally also MNIST) setting?
5) Why is using implicit differentiation for the fixed point of Equation 17 more efficient than directly computing the gradients of the analytical solution, is this due to a diagonal Jacobian? For a second-order univariate polynomial, the exact minimizer is the solution of a univariate linear equation, i.e., a single division, would this not be just as efficient? Can you state the exact equation you use to compute the gradient implicitly? Does this gradient yield different optimization behavior for border extrema?

### Comments
* I would consider removing Figure 5 as the illustrated results seem to be solely based on prior work, as no parameters are optimized or make this point clear in the description.
* I would suggest using equal axis scaling in Figures 1 and 2 (or at least consistent scaling) and the same input bounds for each "column". In its current form, the figure could be interpreted as a direct comparison between linear and quadratic bounds which seems misleading.
* The quantization over all y in Equation 1 seems unintuitive. What is the semantic of $f_N(\mathcal{X})$? Is Equation 1 identical to $\forall y \in \mathcal{Y}, P(y)$? I would consider removing it as later on, only tightness is compared but no properties shown.
* While 3 pages of background are provided on general techniques in NN verification, the background on the polynomial Zonotope representation of Kochdumper & Althoff (2021), perhaps the most relevant for this work, is deferred to the appendix. I would suggest moving it to the main body and highlighting the different representations / sizes between linear and polynomial Zonotopes.

#### Minor Points on Presentation
* Closing bracket is missing in first paragraph of Section 2.1
* The legend in Figure 6 a and b is inconsistent
* "Tightness" is never formally defined
* Semantics of evaluating network on a set in Section 2.2 is not defined

### Conclusion
While the idea of parameterizing the ReLU transformers for polynomial Zonotopes is novel and improves bound tightness, its relevance seems questionable. It neither yields state-of-the-art performance on any task nor propose a novel direction with the potential for significant improvements. Thus, I recommend rejection.
In more detail: The method is extremely limited in scalability, an inherent issue of polynomial Zonotopes that is further exacerbated and seems hard to address. The empirical comparison is lacking key baselines, uses unoptimized re-implementations of (weak) baselines, is limited in scale, and does not show convincing improvements even in the favourably chosen settings.

---

### Official Review · Reviewer_2QHH · 2023-10-30

**Soundness:** 3 good
**Presentation:** 3 good
**Contribution:** 2 fair
**Rating:** 5
**Confidence:** 4

**Summary:**

The paper presents a method for optimising polynomial approximations of ReLUs
in neural network verification. The method relies on maximising the difference
between the ReLU function and its polynomial approximation with respect to a
set of inputs.

**Strengths:**

As far as I am aware this is the first paper that considers optimisable
polynomial relaxations for neural network verification.

**Weaknesses:**

No empirical evaluation is performed on the overhead of the proposed method.

Does not convincingly improve the SoA (optimisable linear approximations often
perform better).

The networks used for evaluation are tiny (the smallest and simplest among the
neural network verification benchmarks). The scalability of the proposed method
to bigger networks and its applicability to a wider range of networks is not
adequately discussed.

Section 2 is unnecessarily large - the section can be shorten significantly as
a low level of detail is not required for the presentation on the novel
contribution in Section 3.

The overall delta with respect to previous work is small.

**Questions:**

See comments above.

---

### Official Review · Reviewer_JLxb · 2023-10-31

**Soundness:** 3 good
**Presentation:** 2 fair
**Contribution:** 2 fair
**Rating:** 3
**Confidence:** 4

**Summary:**

This work is proposes polynomial approximations combined with optimizing the monomial coefficients to calculate sound over approximations of ReLU network behavior. They argue that one can compute polynomial bounds for every neuron by taking any polynomial $p$ and offsetting it with some $\delta = \max \text{ReLU}(x) - p(x)$. With this, they have an upper bound $u(x) = p(x) + \delta$. (Lower bound is analogous). While one could now calculate gradients w.r.t. the monomial coefficients of $p$ binding the ReLU activation using closed form expressions, the authors argue that this is inefficient and they use a method proposed by Blondel et al 2022.

They propose two versions, first $\alpha$-Poly, where optimizable polynomial relaxation with forward propagation are used and second PolyCROWN, which is based on polynomial relaxations in the first layer of the backsubstitution procedure. An anonymized osf-link linking a Julia implementation was provided.

The evaluation is performed on ACAS-XU and MNIST networks from the VNN competition. For ACAS Xu, there are 45 neural networks and for MNIST, 3 fully connected networks, with 2, 4 or 6 layers of 256 neurons each where considered. The authors compare their approach to $\alpha$-Neurify and $\alpha$-Crown and further compare their approach to forward propagation of zonotopes and polynomail zonotopes. First two layers use polynomial relaxations, afterwards linear.

On ACAS Xu, they report improved tightness for PolyCrown with $\alpha$-CROWN. They report that the bounds of PolyCrown are on average 18% tighter for the properties with index >= 100, while it is on average 6% worse on the remaining ones. On MNIST , they again compare PolyCrown with $\alpha$-CROWN for the three networks and 15 images w.r.t. adversarial patches. The authors argue that the tightness compared to $\alpha$-CROWN improves with depth. Improvements seem to be best for patches between 10 and 40 pixels in size.

**Strengths:**

- Convex relaxations are an important for safe and reliable neural networks.
- Very extensive introduction.
- Ideas appear sound. As the implicit differentiation is not further explained, this assesment does not include the application of Blondel et al 2022.

**Weaknesses:**

- The presentation should be improved and the paper should be self contained. Some details remain unclear - i.e. the part about the application of Blondel et al 2022.
- In the abstract it is claimed that polynomial approximations can overcome the loss of precision induced by linear approximations. This appears as an over claim as the two abstract domains appear incomparable for most linear relaxations.
- There is no comparison to something like Beta-CROWN, Multi-Neuron Relaxations, Branch and Bound method:
	- Beta-CROWN: https://arxiv.org/pdf/2103.06624.pdf
	- Branch and Bound: https://arxiv.org/pdf/2205.00263.pdf
	- GCP: https://openreview.net/pdf?id=5haAJAcofjc
- The experiments are not compelling. The results seem to be mixed. There is no comparison to standard certification problems like $\ell_\infty$ robustness.
- Unclear what the limitations of the approach are. Does it work on CIFAR10?
- Unclear on which hardware the experiments where conducted.

Presentation:
- The exact list of contributions that the authors claim remains unclear.
- The paper should be self contained. After equation 17, the authors refer to  Blondel et al 2022, the method (implicit function theorem?) is not further explained.
- The caption in Figure 5 is hard to read.
- Figure 9 would benefit from a logarithmic y scale

**Questions:**

General Questions
- Do higher degree monomials together with optimization give tighter bounds in practice?
- Can one train certifiable networks using this method?
- After equation 13: Could one also replace here with a monotonic polynomial?
- regularization?
- Further, why is the implicit differentiation (Blondel et al 2022) more efficient?

Experiments
- There are in the literature several definitions of tightness (i.e. integral difference or maximum difference) - which one is used here? Is it the sum of the width of the output bounds as in the captions Table 1 or Figure 3?
- Can one achieve state of the art certified accuracy?
- For ACAS Xu - how where the step caps selected?
- Can CIFAR10 networks be handled? What are the limitations of the approach?
- How would this compare to complete verification?
- How would this approach compare to Multi neuron verification/splitting planes?
- On which machine were the experiments run?
- How are the polynomials initialized?
- What does PolyCROWN-opt and $\alpha$-CROWN-opt reffere to?
- How where the examples in Table 1 selected?
- How fast is your reimplementation compared to the original one?

---

### Official Review · Reviewer_PQk4 · 2023-11-01

**Soundness:** 3 good
**Presentation:** 3 good
**Contribution:** 2 fair
**Rating:** 5
**Confidence:** 3

**Summary:**

The paper proposes a new overapproximation method for the ReLU activation function with a new designed trainable nonlinear abstraction. The proposed abstract domain allows tracing the overapproximated output interval of neural network back to the input layer directly through symbolic computation and optimizing the over approximation of the ReLU activation function. The improved performance is demonstrated in the experimental section when comparing to the state-of-the-art neural network verification methods.

**Strengths:**

- The paper is sound and the topic of the paper is of high interest to the research community.
- The paper does a good job at introducing technical details and makes the paper easy to follow.
- The empirical study shows improvement of the proposed method over existing neural network verification methods.

**Weaknesses:**

- The main concern I have is that how costly is the optimization of the parameterized polynomial overapproximation. The optimization not only needs to do per unstable neuron but also for each input set. In the experiment, the models are having relatively small numbers of neurons and only one input set for each property.
- Though verification time improvement was shown compared with other approaches, it is not clear whether that includes the optimization of the parameters in the polynomial function overapproxiamtions.

**Questions:**

I noticed the experiments are done on only small datasets (e.g. ACAS Xu and MNIST). What is the main bottleneck from testing the proposed method on large-scale datasets/models (e.g. ImageNet)? Is it the model size? The input dimension? The limitations of verification methods for large models? It would be good if the authors can comment on this in the paper and discuss the limitations of the proposed method.

---

### Author Response · Authors · 2023-11-23
**General Response**

We would like to thank the reviewers for their detailed analyses and helpful comments.
Some issues were brought up by multiple reviewers, which we would like to address first.

## Comparison with branch-and-bound (BaB) / multi-neuron relaxation

Our proposed approach is a bound propagation method with optimizable parameterized ReLU relaxations. We chose to compare our approach to $\alpha$-CROWN as this is – to our knowledge – still the best bound propagation method.
Moreover $\alpha$-CROWN is very similar to our approach as it also optimizes parameterized relaxations (albeit linear relaxations instead of our polynomial relaxations).

BaB and multi-neuron relaxation based approaches usually add branching or cutting planes on top of the results of bound propagation methods ($\alpha$-$\beta$-CROWN and GCP-CROWN still use $\alpha$-CROWN as a component, PRIMA or MN-BaB work on top of the similar DeepPoly method). We therefore consider our approach orthogonal to BaB and multi-neuron relaxation based methods; it could be included in existing BaB algorithms (easily so for input splitting; for branching on neuron state, further research would be required).

## Scalability

It is clear that propagating polynomials through neural networks has a higher time complexity than propagation of linear functions.
However, the limitations in scalability seem to be exaggerated in our experiments by our implementation.

### Implementation

Upon the comments of the reviewers, we looked into the performance of our code and were able to achieve a 2-3x speedup for our implementation of $\alpha$-CROWN and 3-5x speedup for our implementation of PolyCROWN on preliminary experiments on a standard notebook.
We note that we did not change details of our approach or include pruning of fixed neurons - the improvements were only due to handwritten differentiation rules and better type stability allowing Julia to generate efficient code.
We reran the experiments and will include the results in the final version of our paper.
Additionally, we discuss preliminary results in the next response.

### Theoretical Considerations

Considering a feedforward network with $m$ (unfixed) inputs and $L$ layers of $n$ neurons each, we can calculate the time complexities of the backsubstitution-based algorithms as
- $O(Ln^2m + L^2n^3)$ for bounds computation via CROWN and
- $O(Ln^2m^d + L^2n^3)$ for PolyCROWN with a polynomial relaxation of fixed degree $d$ at the first layer.
Our approach will therefore not scale to neural networks operating on images, when a large number of pixels is unfixed (as is the case in verifying $L_\infty$ robustness).
Scalability to deeper or wider networks, however, is the same as for CROWN.

Time complexity for the forward propagation-based algorithms is given by
- $O(Ln^2m)$ for Neurify and
- $O(Ln^2m^{d^L})$ for forward propagation with polynomial relaxation of fixed degree $d$ at each layer

This last variant clearly scales badly with both the number of unfixed inputs as well as the depth of the network.
Therefore, the number of monomials is truncated to a constant number $c$ of the monomials whose corresponding monomial coefficient vector has the largest $L_2$ norm before every propagation through a polynomial relaxation. The remaining monomials are overapproximated via interval arithmetic and the bounds are added to the constant term of the polynomial.
This fact was only mentioned in the appendix and we admit that it should be included in the main paper.
Then, we obtain
- $O(Lc^d (d \cdot \log(c) + n^2))$ for forward propagation with polynomial relaxation of fixed degree $d$ at every layer with truncation

---

> ### Author Response · Authors · 2023-11-23
> **Addressing that PolyCROWN does not consistently beat $\alpha$-CROWN**
>
> Although the graph in Figure 4a may seem inconsistent, we remark that PolyCROWN was able to calculate tighter bounds than $\alpha$-CROWN in 122 out of 186 instances for ACAS-Xu.
> PolyCROWN performed especially well in cases, where bounds computed by $\alpha$-CROWN were relatively loose as can be seen on the right of figures 4a and 4b.
>
> The experiments on the larger MNIST networks indeed show mixed results.
> We believe that an important reason for this is the lower iteration speed of PolyCROWN compared to $\alpha$-CROWN.
> In all of the five cases we ran with a one hour timeout, PolyCROWN is able to compute tighter bounds for the majority of optimization steps than $\alpha$-CROWN as can be seen in Figure 9 in the appendix. Additionally, for three of these optimization runs, $\alpha$-CROWN seems to have converged, while PolyCROWN might still be able to improve bounds tightness, if we increase the timeout.
> This is the case, even though we selected the five instances where PolyCROWN fared worst compared to $\alpha$-CROWN, measured in relative improvement over the experiments with 5 minute timeout.
>
> ## Rerun with more efficient Implementation
>
> Our claim is supported now that we reran the experiments with our more efficient implementations:
> PolyCROWN now improves upon our implementation of $\alpha$-CROWN by calculating tighter bounds for 102/135 MNIST instances with a 5min timeout. With the bounds being on average 1.4x tighter.
>
> The results for ACAS-Xu did not change notably.

---

### Meta-Review · Area_Chair_UHH8 · 2023-12-07

**Metareview:**

Summary: The submission proposes the use of polynomial approximations of non-linear activations for neural network verification. Inspired by the parametric optimization of linear relaxations, it proposes a similar parametric optimization of the polynomial approximations.

+ The paper addresses a relevant problem, namely, neural network verification.
+ The method is technically sound.

- The experimental comparison shows marginal improvements.
- Appropriate baselines have not been thoroughly tested.

**Justification For Why Not Higher Score:**

- The experimental comparison shows marginal improvements.
- Appropriate baselines have not been thoroughly tested.

For the submission to be accepted at a top-tier ML venue, the experiments would need to be significantly improved, as detailed in the reviewers' comments.

**Justification For Why Not Lower Score:**

N/A

---

### Decision · Program_Chairs · 2024-01-16

Reject